# Fair and Private Approximate Kernel Ridge Regression

## Abstract

Even though Kernel Ridge Regression (KRR) is a well-studied nonparametric problem, its applicability is hindered by its time and space complexity. One of the most common methods for circumventing this issue is the Nyström approximation method, which approximates the full kernel matrix by a low rank matrix. The implementation of approximate KRR for real-world systems, such as medical data, further raises concerns regarding protection of privacy of the patients, as well as ensuring patients are treated fairly, irrespective of their background. In this paper, we study the problem of private and fair KRR that is scalable. To the best of our knowledge, this is the first work that considers the privacy and fairness aspects of approximate KRR. Building upon well-known techniques in Nyström approximation and differential privacy, we propose a technique for computing the Nyström approximation in such a way each demographic group is represented in the basis. We also ensure that the landmarks do not reveal any information regarding the private dataset. Further, the coefficients of the KRR problem are learned in a privacy preserving manner. We compare different variations of this framework empirically on a set of real-world datasets.

## 1 Introduction

Kernel Ridge Regression (KRR) has proved to be one of the most elegant algorithms when dealing with nonlinear complicated data. It admits a closed form solution which has enabled extensive analysis of the learned models. Learning the closed form solution, however, can become computationally expensive as the time complexity scales as $O(n^3)$ and the space complexity scales as $\theta(n^2)$, where $n$ is the size of the dataset. This prohibits its usage for large scale databases. Another issue of these methods is that the learned coefficients are deterministic nonlinear transformations of the training dataset itself, which leaks sensitive information about the individuals in the dataset.

To address the scalability issue of KRR, various methods such as random features and Nyström approximation have been used. Nyström approximation Williams & Seeger (2000), in particular, is a data dependent method that samples points (*landmarks*) from the training data to compute a low rank approximation of the full kernel matrix. Several sampling strategies have been proposed for selecting the landmarks Kumar et al. (2012) such as cluster centers, uniform sampling, and ridge leverage score (RLS) sampling Alaoui & Mahoney (2015). The RLS sampling methods have been widely studied as they provide an analogue to leverage score sampling used in approximate ordinary least square solutions. The use of such sampling methods for computing the approximate kernel matrix raises the concern of representation bias. If the dataset is known to contain data from various demographic groups, with one group being the majority, then the sampling methods could ignore or undersample points from the minority groups. A KRR model trained on such an approximation will inevitably be biased against the minority group.

In order to address these issues, in this paper, we study the problem of fair and differentially private kernel ridge regression. We propose a novel sampling method for computing the Nyström approximation, such that the closed form solution is differentially private and fair. We consider the notion of differential privacy (DP) and the minimax notion of fairness. While there have been works on differentially private kernel learning Chaudhuri et al. (2011); Kifer et al. (2012) and fair kernel learning Okray et al. (2019); Pérez-Suay et al. (2017), to the best of our knowledge, this is the first work that explores KRR with privacy and fairness constraints simultaneously. Consider an example where a pharmaceutical company wants to decide who gets

access to a new drug based on their predicted success probability. In such a scenario, the privacy of the patients' information must not be compromised. At the same time, it is important that the model does not discriminate against elderly patients, or patients from minority groups. Even though the individuals' information are private, their protected attribute such as gender or age are important for making the decisions at test time. Since the number of patients could be huge, an approximate model is required.

In our work, we assume that we have access to a small public validation dataset which is sampled from the same distribution as the private training dataset. Similar methods have been used for various data analysis tasks without affecting the privacy budget Lowy et al. (2024).

## 2 Preliminaries

Here, we briefly discuss the necessary background on scalable kernel learning, kernel ridge leverage scores, minimax group fairness, and pure differential privacy. The private dataset is represented as $D^{priv} = \{x_i, a_i, y_i\}_{i=1}^n$ where $x_i \in \mathbb{R}^d$ are the features, $y_i \in \mathbb{R}$ are output values and $a_i \in \{1, \ldots, m\}$ represent the group information. We also have access to a small public dataset $D^{pub}$ of size $r \ll n$.

### 2.1 Kernel Ridge Regression

The Kernel ridge regression (KRR) problem Schölkopf et al. (2001); Wahba (1990) solves

$$\min_{f \in \mathcal{H}} \frac{1}{n} \sum_{i=1}^n (f(x_i) - y_i)^2 + \frac{\lambda}{n} \|f\|_{\mathcal{H}}^2, \tag{1}$$

where $\mathcal{H}$ is a Reproducing Kernel Hilbert Space (RKHS) induced by the kernel $k$ and $\lambda > 0$ is a regularization parameter. The minimizer of Problem 1 admits the form $f^*(\cdot) = \sum_{i=1}^n \alpha_i k(x_i, \cdot)$,

where $\alpha = (K + \lambda I)^{-1} y$. Computing these coefficients $\alpha$ is a computationally demanding step, which has led to the development of various approximation algorithms.

### 2.2 Nyström Method

Let $k : \mathbb{R}^d \times \mathbb{R}^d \to \mathbb{R}$ be a positive semidefinite kernel with $K \in \mathbb{R}^{n \times n}$ begin the associated kernel matrix for the dataset, where $K_{ij} = k(x_i, x_j)$. Storing the kernel matrix requires $\theta(n^2)$ space and many kernel based methods require the computation of its inverse, which takes $O(n^3)$ time. To address these problems, the Nyström approximation method was developed. This method approximates $K$ using a low-rank sketch $\tilde{K}$ Williams & Seeger (2000); Drineas et al. (2005); Gittens & Mahoney (2013); Alaoui & Mahoney (2015) such that $\tilde{K}$ is spectrally close to $K$.

Computing a rank $t$ Nyström approximation requires a set of $t \ll n$ Nyström basis vectors. For computing this basis, a set of $t$ points are computed from the dataset. These $t$ points could either be sampled from the dataset or represent certain aggregations of the dataset, such as cluster centers. The Nyström approximation is then computed as

$$\tilde{K} = K_{tn}^\top K_{tt}^\dagger K_{tn}, \tag{2}$$

where $K_{tn}$ is the kernel matrix consisting of the kernel values between the landmarks and the dataset, $K_{tt}$ is the kernel matrix of the landmarks, and $\dagger$ is the pseudo-inverse. Given $\tilde{K}$, the Nyström mapping for a point $x \in \mathbb{R}^d$ is given as

$$\Phi_{nys}(x) = K_{tt}^{-1/2} \begin{pmatrix} k(x, x_{z_1}) \\ \vdots \\ k(x, x_{z_t}) \end{pmatrix} \in \mathbb{R}^t \tag{3}$$

where $\{z_1, \ldots, z_t\}$ are the landmarks. Nyström approximation reduces the complexity of kernel learning with time complexity $O(nt^2)$ and space complexity of $O(t^2)$.

**Approximate Kernel Ridge Regression** Using the Nyström mappings computed in 3, the KRR objective can be stated as

$$\min_{\alpha \in \mathbb{R}^t} \mathcal{L}(\alpha) = \frac{1}{n} \sum_{i=1}^{n} (\alpha^\top \Phi_{nys}(x_i) - y_i)^2 + \frac{\lambda}{n} \|\alpha\|_2^2, \tag{4}$$

The coefficients are computed as

$$\alpha = (\Phi_{nys}^\top \Phi_{nys} + \lambda I)^{-1} \Phi_{nys}^\top y \tag{5}$$

**Kernel Ridge Leverage Scores** In order to obtain statistically efficient Nyström approximations, the landmarks can be sampled according to the kernel ridge leverage scores (RLS) Alaoui & Mahoney (2015); Cohen et al. (2017); Avron et al. (2017), which are defined as

$$\tau_i = (K(K + n\lambda I)^{-1})_{ii}. \tag{6}$$

The RLS of a point indicates its influence on statistical problems involving the dataset. Points which "stand out" are assigned higher leverage scores than the points lying in low-dimensional subspaces of the dataset.

### 2.3 Differential Privacy

Differential privacy Dwork et al. (2006); Dwork & Roth (2014) has been widely used for the development of various private algorithms that protect against the leakage of any individual's private information.

**Definition 1** *Let $D$ and $D'$ be databases that differ in one data point. A randomized algorithm $\mathcal{M}$ is said to be $(\varepsilon, \delta)$-differentially private ($(\varepsilon, \delta)$-DP) if*

$$\Pr[\mathcal{M}(D) \in \mathcal{S}] \leq e^\varepsilon \Pr[\mathcal{M}(D') \in \mathcal{S}] + \delta \tag{7}$$

*where $\mathcal{S}$ is the output space of $\mathcal{M}$.*

The Gaussian mechanism is $(\varepsilon, \delta)$- differentially private. where $h(.)$ is some function and $\Delta$ is the $\ell_2$ sensitivity of $h$.

## 3 Related Work

### 3.1 Kernel Ridge Regression and Scalable Nyström Approximation

Kernel methods enable nonlinear learning in reproducing kernel Hilbert spaces (RKHS) through regularized empirical risk minimization (ERM). The generalized representer theorem Schölkopf et al. (2001) characterizes the finite-dimensional structure of optimal solutions, while classical spline theory Wahba (1990) provides the statistical foundation for kernel regularization. Kernel ridge regression (KRR) is a canonical instance, but exact training requires solving a dense $n \times n$ linear system, incurring $O(n^3)$ time and $O(n^2)$ memory, which is prohibitive at database scale.

The Nyström method Williams & Seeger (2000) reduces computational complexity by constructing a low-rank approximation of the kernel matrix via landmark sampling. Drineas et al. (2005) provided early theoretical analysis of Gram matrix approximation for kernel-based learning, and Gittens and Mahoney Gittens & Mahoney (2013) refined spectral error guarantees. Alaoui and Mahoney Alaoui & Mahoney (2015) established that ridge leverage score (RLS) sampling yields statistically optimal approximations for KRR, with rank proportional to the effective dimension. Cohen et al. (2017) developed input-sparsity-time algorithms for

approximating ridge leverage scores, enabling near-linear preprocessing. Random Fourier feature approximations with statistical guarantees were analyzed in Avron et al. (2017), and structured landmark selection via kernel $k$-means++ was studied in Oglic & Gärtner (2017).

These advances provide the algorithmic primitives necessary for scalable KRR: subquadratic memory, controllable statistical error via effective dimension, and compatibility with matrix sketching pipelines. Our work builds on RLS-based Nyström approximation to obtain statistically grounded and computationally efficient kernel regression under additional fairness and privacy constraints.

### 3.2 Fairness in Kernel Learning and Minimax Group Objectives

Fairness in supervised learning has been formulated either as constrained ERM or as distributionally robust optimization (DRO). Donini et al. (2018) studied ERM under fairness constraints with generalization guarantees. In kernel settings, fairness has been incorporated via modified kernels or feature embeddings (Pérez-Suay et al., 2017; Okray et al., 2019). Fair clustering under group constraints was analyzed in (Bera et al., 2019).

A complementary perspective formulates fairness as worst-group robustness. Hashimoto et al. (2018) proposed minimizing the maximum loss over subpopulations in repeated deployment settings. Sagawa et al. (2019) demonstrated that distributionally robust training improves worst-group generalization, while Duchi & Namkoong (2021) provided statistical guarantees for uniform group performance under DRO formulations. Diana et al. (2021) further developed explicit minimax group fairness algorithms with empirical evaluation, directly optimizing worst-group risk.

These works establish minimax group risk as a principled fairness objective. Recently, Lee et al. (2024) proposed using DP Nyström random features for the purpose of general kernel learning. Our problem formulation aims to minimize the worst case group loss for DP KRR while preserving computational scalability.

### 3.3 Differentially Private Empirical Risk Minimization

Differential privacy (DP) was introduced in Dwork et al. (2006) and systematically developed in (Dwork & Roth, 2014). For convex ERM, Chaudhuri & Monteleoni (2008); Chaudhuri et al. (2011) proposed objective and output perturbation mechanisms with utility guarantees under strong convexity. Kifer et al. (2012) extended these results to high-dimensional convex ERM. Hall et al. (2013) analyzed differential privacy for function-valued outputs and RKHS settings, demonstrating applicability to kernel methods. Sampling-based improvements to privacy–utility trade-offs were studied in (Ebadi et al., 2016). These results imply that strongly convex regularized ERM, including KRR, admits principled pure $\varepsilon$-DP guarantees via objective perturbation.

### 3.4 Fairness under Differential Privacy

The interaction between fairness and privacy has received increasing attention. Jagielski et al. (2019) proposed algorithms for differentially private fair learning. Tran et al. (2021a;b) studied ERM and deep learning under fairness constraints with privacy guarantees. Ding et al. (2020) introduced calibrated functional mechanisms for private and fair classification. Bagdasaryan et al. (2019) showed that differential privacy can induce disparate impact across demographic groups, revealing inherent tension between privacy and fairness.

Existing work in this space primarily addresses classification or deep neural networks and does not consider scalable kernel learning. Moreover, prior analyses treat fairness constraints and privacy mechanisms largely independently, without jointly analyzing approximation error from kernel sketching.

## 4 Our Method

Given a dataset $(X, A, Y) = \{x_i, a_i, y_i\}_{i=1}^{n}$ where $x_i \in \mathbb{R}^d, y_i \in \mathbb{R}$, and $a_i \in \{1, \ldots, m\}$, where $m$ is the number of demographic groups, we aim to learn the coefficients of the KRR problem, $\alpha_{priv}$, such that no

information about any individual in the training data is revealed, and simultaneously, maximum cost incurred for any of the demographic groups, as well as the disparity among the groups is minimized.

The Nyström approximation requires the computation of the $K_{tt}$ matrix, the kernel matrix of the landmark points. Sampling points from the private dataset for computing $K_{tt}$ would lead to privacy leakage. Lee et al. (2024) bypassed this problem by computing Nyström random features from using the cluster centers of a DP $t$-means clustering as landmarks. While this preserves the privacy of the datapoints, it can adversely affect the fairness of the model. Firstly, if the datapoints of the majority group are densely located in the feature space, and the minority group is sparse, then the cluster centers would be placed in the dense clusters to minimize the clustering cost. This, in turn, causes the minority groups to be underrepresented in the Nyström approximation, which also degrades the performance of the downstream KRR on such groups. Secondly, since the cluster centers have to be computed in a DP preserving manner, for a strict privacy budget (small $\varepsilon$), the amount of noise added for privacy could overwhelm the computed centers for very small groups. In this case, even though the cluster centers are private, their utility could be diminished.

An obvious solution is to allocate the number of landmarks to each group proportional to their size, and then compute the cluster centers using the DP clustering. Even in this case, the cluster centers might be too noisy (due to privacy-preserving noise) to cover the minority groups. Another possible solution is to allocate the number of landmarks based on the variance of each group, with densely clustered groups being allocated fewer number of clusters. Even in this case, the small size of the minority groups could cause the DP noise to overwhelm the computation of cluster centers. This problem will persist whenever the group sizes are small, which negatively impacts the utility of the learned model.

To alleviate this issue, we propose utilizing the auxiliary public dataset, $\mathcal{D}^{pub}$ for sampling the landmarks. This approach follows recent works Lowy et al. (2024) that utilize the public dataset to preserve the privacy budget. We assume that $\mathcal{D}^{pub}$ has been sampled from the same data distribution as the private training dataset $D^{priv}$, such that the proportion of each group in $\mathcal{D}^{pub}$ is same as that in $\mathcal{D}^{priv}$. The size of $\mathcal{D}^{pub}$ is $r \ll n$. The pseudo-code for our proposed method is given in Algorithm 1.

---

**Algorithm 1** Fair and Private Kernel Ridge Regression (FPKRR)

---

**Require:** Private dataset $\mathcal{D}^{priv} : \{x_i, a_i, y_i\}_{i=1}^{n}$, public data $\mathcal{D}^{pub} : \{x_j, a_j, y_j\}_{j=1}^{r}$, privacy budget : $\varepsilon$, $\delta = 1/n^2$, rank of Nyström approximation : $t$, kernel : $k$

**Ensure:** KRR coefficients, $\alpha_{priv}$

  1: Sample landmarks $\ell = \{\ell_1, \ldots, \ell_m\}$ for each group from $\mathcal{D}^{pub}$, where $|\ell_j| = t_j$ and $\sum_{j=1}^{m} t_j = t$.

  2: Compute the Nyström mappings $\Phi_{nys}(\cdot) = [k(z_1, \cdot), \ldots, k(z_t, \cdot)]$

  3: Compute $\alpha_{priv}$ as in Equation 9.

    **Return:** $\alpha_{priv}$

---

We sample the Nyström landmarks using the kernel ridge leverage scores (RLS). Even though computing the exact RLS distribution incurs cubic time complexity in the size of the dataset, in our case, we need only to compute these scores for the $r$ points in $\mathcal{D}^{pub}$ which is much less than $n$. Naively sampling according to the RLS might still lead to the minority groups being underrepresented in the Nyström approximation. We alleviate this issue by a group-wise stratified sampling strategy.

## 4.1 Stratified RLS Sampling

The RLS assigns weights to points based on their influence on some statistical query. Points that lie in a low-dimensional subspace of the dataset generally have low leverage scores, as they are not influential enough to impact the results of the statistical query. From the perspective of fairness, if the points of a minority group lie in this low-dimensional region, then the probability of sampling a landmark point representing that group is small. Consequently, even though the overall model will have a good utility, the utility for that particular group will be adversely affected, since there is no Nyström basis vector representing that group. Distributing the landmarks equally among the groups is also not advisable as for small groups, it could be

the case that the number of landmarks is greater than the group size. In such a scenario, the Nyström landmarks for that group will get repeated, which in turn will make the Nyström approximation low-rank or highly ill-conditioned.

To address these issues, we assign the number of landmarks to each group based on its importance measured as its contribution to the leverage scores. In particular, for a group $g \in [m]$, its importance or "leverage mass" is defined as $\rho_g = \frac{\sum\limits_{i:a_i=g} \tau_i}{\sum\limits_{j \in [n]} \tau_j}$. Based on $\rho_g$, the number of landmarks assigned to group $g$ is $t_g = t\rho_g$. For every group, we sample the landmarks from the points of that group using their leverage scores. That is, for a point $i$ in group $g$, the probability of it being sampled is $p_i = \frac{\tau_i}{\sum\limits_{i:a_j=g} \tau_j}$.

## 4.2   Differentially Private Kernel Ridge Regression

Once the Nyström mappings have been computed, we optimize the KRR objective such that computing the optimal coefficients is differentially private. For this purpose, we apply the objective perturbation approach of (Kifer et al., 2012). Consider a kernel $k(\cdot, \cdot)$ such that the convex loss function $\ell(\alpha) = ||\Phi_{nys}(X)\alpha - y||_2^2$ has bounded gradient $\ell_2$ norm as $||\nabla\ell(\alpha)||_2 \leq \zeta$. For DP parameters $(\varepsilon, \delta)$, let $b \in \mathbb{R}^t$ be a random vector sampled as $b \sim \mathcal{N}(0, \frac{\zeta^2(8\log(2/\delta)+4\varepsilon)}{\varepsilon^2} I_{t \times t})$. Then, the differentially private KRR objective can be stated as

$$\arg \min_{\alpha \in \mathbb{R}^t} \mathcal{L}_{pert}(\alpha) = \frac{1}{n}||\Phi_{nys}(X)\alpha - y||_2^2 + \frac{\lambda}{n}||\alpha||_2^2 + \frac{b^\top \alpha}{n}. \tag{8}$$

The optimal coefficients can then be computed as

$$\alpha_{priv} = (\Phi_{nys}(X)^\top \Phi_{nys}(X) + \lambda I)^{-1}(\Phi_{nys}(X)^\top y - \frac{b}{2}). \tag{9}$$

In this work, we want to ensure that the solution to the approximate KRR is minimax fair, that is, the maximum group loss is minimized, and also, the disparity is minimized. Incorporating these constraints along with the objective perturbation for privacy will result in a minimax optimization problem. We leave such a formulation as a future work. Here, we consider only the objective perturbation method for enforcing privacy, with the stratified RLS sampling ensuring fair representation of each group in the Nyström approximation. We consider two ways of optimizing the KRR objective such that the maximum group loss as well as disparities are reduced.

**Decoupling**   The first method we consider is a *decoupling* approach - learn a separate set of coefficients for each group independently. At inference, we assume that the sensitive attribute of the test point is known publicly. This is reasonable for applications such as in the medical domain, where the age and gender of the patients is required. In Dwork et al. (2018), it was shown that whenever it is legal to use the sensitive attributes for machine learning systems, learning decoupled group-wise classifiers can result in models that are fair across groups. We adopt this approach for privately learning the KRR coefficients separately for each group. Since, the datapoints for various groups are mutually exclusive, the privacy budget $\varepsilon$ need not be split among the groups. However, if a group size is small, i.e., then the regularization term in 8 becomes large due to which the utility for the group could decrease. In order to mitigate this issue, we adopt a different approach.

**Feature Augmentation**   Instead of learning the coefficients in a decoupled manner, we adapt the feature augmentation approach of Daumé III (2007) to our problem. We learn a set of *global* coefficients that minimize the errors on the full dataset along with a set of *local* group-wise coefficients that minimize group-wise errors.

Let $X_g$ be the set of points belonging to group $g$ of size $n_g$, and let $\phi_g = \phi(X_g)$ be the $n_g \times t$ matrix of the Nyström mappings of $X_g$. Let $\Phi$ be a $n \times (m+1)t$ block matrix where the $g$th block corresponds

to group $g$, that is $\Phi_{aug} = \begin{bmatrix} \Phi_1 \\ \vdots \\ \Phi_m \end{bmatrix}$. Here, the augmented features for group $g$ is represented by $\Phi_g = \frac{1}{\sqrt{2}}[\phi_g, 0, \ldots, \phi_g, 0, \ldots, 0]$ which is a $n_g \times (m+1)t$ matrix. The first $t$ columns of $\Phi_{aug}$ are filled by all of the groups, while the remaining block of non-zero columns corresponds to the group numbers. For example, if $G = 3$, then $\Phi_{aug} = \frac{1}{\sqrt{2}} \begin{bmatrix} \phi_1 & \phi_1 & 0 & 0 \\ \phi_2 & 0 & \phi_2 & 0 \\ \phi_3 & 0 & 0 & \phi_3 \end{bmatrix}$. The factor of $\frac{1}{\sqrt{2}}$ normalizes each row such that the row norms do not increase due to the feature augmentation. Given this augmented feature matrix, the downstream KRR coefficients can be grouped as $\alpha = [\alpha_{global}, \alpha_1, \ldots, \alpha_m]$ where $\alpha_{global}$ is the set of coefficients that learn the global structure while $\alpha_g$ learns the coefficients for group $g$. We note here the augmented feature matrix is not released, only the private coefficients $\alpha_{priv} = [\alpha_{global}^{priv}, \alpha_1^{priv}, \ldots, \alpha_m^{priv}]$ are released. Thus, an adversary cannot figure out the group information of any individual in $\mathcal{D}^{priv}$. We also note here that during inference, the group information must be provided by the user so that the appropriate weights $[\alpha_{global}, \alpha_g]$ can applied. Even though this is not a standard practice in most legal and ethical systems, certain domains such as medical requires group information such as gender or age. Our proposed method is applicable to such systems in the real world.

## 5 Experiments

Here, we experiment on two real-world datasets and evaluate the impact of using the stratified RLS sampling followed by optimizing a DP KRR objective. The first dataset that we experiment on is the *Law School (LAW)* dataset which consists of records for $20,649$ students, with GPA as the target and race as the sensitive attribute, $m = 4$ (Wightman, 1998). This dataset is very imbalanced with the percentage of white students being $85\%$, hispanics being $5\%$, blacks being $6\%$ and Asian students being $4\%$. The second dataset that we consider is the *Medical Expenditure Panel Survey (MEPS)* dataset (2015) consisting of the records for $35427$ patients, with the total healthcare expenditure as the target, and gender as the sensitive attribute, $m = 2$ [1]. This dataset is much more balanced with $52\%$ being male patients and $48\%$ begin females.

For KRR, we considered the RBF kernel since $k(\cdot, \cdot) \leq 1$, and it also leads to the eigenvalues of the Hessian of the loss function be bounded, the allowing the bounding the DP noise scale. This may not be the case for other kernels such as the polynomial or linear kernels. For computing the kernel RLS (6), we use $\lambda = 1e-4$. We split the dataset as $75\%$ for training ($\mathcal{D}^{priv}$), $10\%$ for validation ($\mathcal{D}^{pub}$), and $15\%$ for testing.

For computing the Nyström mappings, $\Phi_{nys}$, we consider two settings : $\Phi_{nys}^{RLS}$ : computed with landmarks sampled from $\mathcal{D}^{pub}$ using kernel RLS, and $\Phi_{nys}^{stratified}$ : computed with landmarks sampled from $\mathcal{D}^{pub}$ using stratified RLS sampling. For optimizing the DP KRR objective, we compare 5 different combinations :

1. $\Phi_{nys}^{RLS} + DPKRR$ : Compute Nyström mappings as $\Phi_{nys}^{RLS}$, and optimize a DP-KRR objective

2. $\Phi_{nys}^{stratified} + DPKRR$ : Compute the Nyström mappings as $\Phi_{nys}^{stratified}$, and optimize a DPKRR objective

3. $\Phi_{nys}^{stratified} + Decoupled$ : Compute the Nyström mappings as $\Phi_{nys}^{stratified}$, and optimize a DPKRR objective for each group independently (similar to Dwork et al. (2018))

4. $\Phi_{nys}^{stratified} + FA$ : Compute the Nyström mappings as $\Phi_{nys}^{stratified}$, and optimize a DPKRR objective using the feature augmentation method (Section 4.2)

5. DPNyström : Compute the Nyström mappings using the DP-Kmeans method proposed in Lee et al. (2024) and then optimize the DPKRR objective.

---

[1] https://meps.ahrq.gov/mepsweb/data_stats/download_data_files_detail.jsp?cboPufNumber=HC-181

For each of these methods, we compare two fairness metrics : the maximum group loss $\max_{g\in[G]} L_g$ and the maximum groupwise loss disparity $\Delta = \max_{g\in[G]} L_g - \min_{g\in[G]} L_g$. We want both of these metrics to be lower. The experiments were run independently for 10 different seeds and the aggregate metrics have been reported.

## 6 Results and Discussions

In Tables 1 and 2, we compare the maximum disparity and maximum group loss for the two datasets across various methods. The results displayed in the tables are for $t = 50$ landmark points, with the privacy budget $\varepsilon = 1.0$ and $\delta = \frac{1}{n^2}$.

Table 1: Comparison of Max Disparity ($\downarrow$) and Max Group Loss ($\downarrow$) for the *LAW* dataset.

| **Metric** | $\Phi_{nys}^{RLS}$ + DPKRR | $\Phi_{nys}^{stratified}$ + DPKRR | $\Phi_{nys}^{stratified}$ + Decoupled | $\Phi_{nys}^{stratified}$ + FA | DPNyström |
|---|---|---|---|---|---|
| Max Disparity | $0.15 \pm 0.16$ | $\mathbf{0.11 \pm 0.03}$ | $0.27 \pm 0.31$ | $0.13 \pm 0.03$ | $5.38 \pm 0.96$ |
| Max Group Loss | $0.32 \pm 0.19$ | $\mathbf{0.27 \pm 0.03}$ | $0.42 \pm 0.30$ | $0.29 \pm 0.03$ | $10.56 \pm 0.19$ |

In both of the tables, it can be observed that in most of the cases the lowest disparity is obtained when the Nyström mappings are computed as $\Phi_{nys}^{stratified}$. In Table 1, the minimum disparity is observed for the $\Phi_{nys}^{stratified} + DPKRR$ method, where the stratified RLS sampling captures the important points from each protected group. It is lower than $\Phi_{nys}^{stratified} + FA$ which might be explained by the fact that the dimension of the noisy vector $b$ is $5t$ compared to just $t$ for the case of $DPKRR$. This is not observed in Table 2 because both of the groups are approximately of the same size, so the noise does not have much affect on the maximum group loss.

When the DPKRR objective is optimized in a decoupled manner for each group, in Table 1 even though the group wise error may be reduced, since some of the groups are much smaller, disparity and the loss are much higher. Another reason is that the noise vector is divided by the group size, causing stronger regularization for smaller protected groups (since the regularization term is $b/n_g$ with $n_g$ being the size of group $g$.). This is not observed in Table 2 because both of the groups are large enough such that the noise does not have a large impact.

Table 2: Comparison of Max Disparity ($\downarrow$) and Max Group Loss ($\downarrow$) for the *MEPS* dataset.

| **Metric** | $\Phi_{nys}^{RLS}$ + DPKRR | $\Phi_{nys}^{stratified}$ + DPKRR | $\Phi_{nys}^{stratified}$ + Decoupled | $\Phi_{nys}^{stratified}$ + FA | DPNyström |
|---|---|---|---|---|---|
| Max Disparity | $0.97 \pm 0.32$ | $0.93 \pm 0.40$ | $\mathbf{0.89 \pm 0.48}$ | $\mathbf{0.89 \pm 0.48}$ | $8.41 \pm 1.64$ |
| Max Group Loss | $9.25 \pm 0.21$ | $9.23 \pm 0.40$ | $\mathbf{9.01 \pm 0.44}$ | $\mathbf{9.01 \pm 0.43}$ | $46.71 \pm 1.63$ |

In Figures 1 and 2, we plot the maximum disparities as well as the maximum group losses for $\varepsilon = 1.0$ for varying number of landmarks. For the *LAW* dataset, it can be observed that both of these quantities increase with the increase in the number of Nyström basis vectors. This is because the worst case group loss for this dataset is incurred by one of the minority groups. Also, the noise vector $b$ has dimensionality $= t$ or $5t$(for the feature augmentation method) which causes the noise to increase as the number of landmarks increases. For the small groups, this noise overwhelms the contributions of the points in that group, due to which both the disparity as well as the maximum group loss degrades.

On the other hand, for the *MEPS* dataset, both the disparities and the maximum group losses decreases with the increase in the number of Nyström landmarks. The group sizes for this dataset are balanced and large enough, due to which as noise added to preserve privacy does not overwhelm the contribution of the points.

In Figures 3 and 4, we plot the disparities as well as the maximum group losses for $t = 150$ landmarks while varying the privacy budget. For the *LAW* dataset, it can be observed that as the privacy constraints are

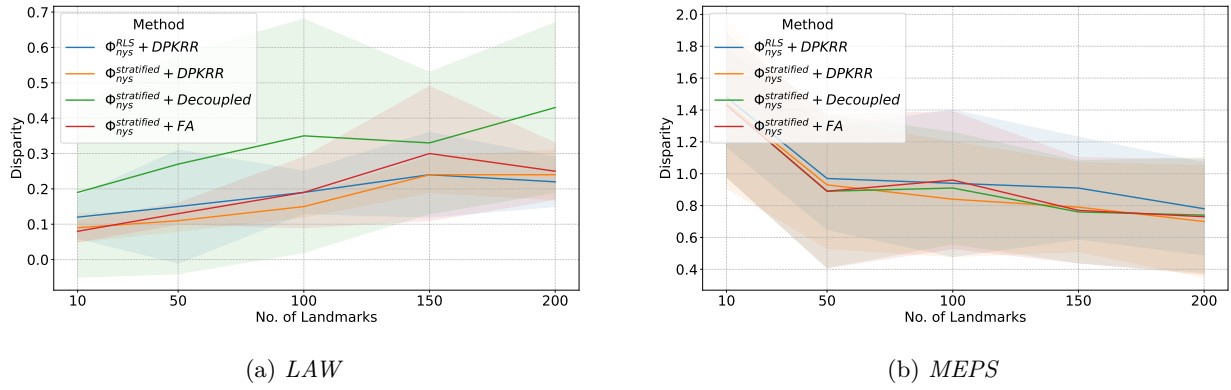

(a) *LAW*

(b) *MEPS*

Figure 1: Comparison of Maximum Disparity for $\varepsilon = 1.0$.

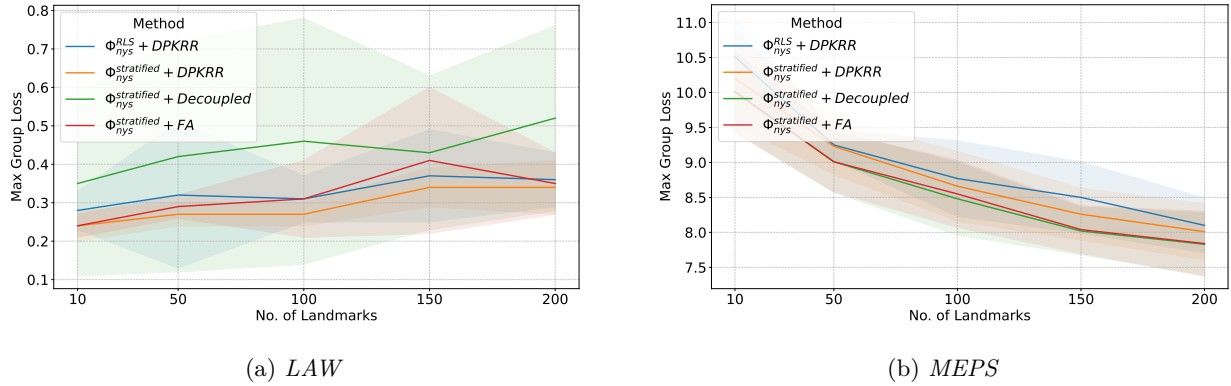

(a) *LAW*

(b) *MEPS*

Figure 2: Comparison of the Maximum Group Loss for $\varepsilon = 1.0$.

loosened, the disparities as well as the maximum group losses decrease. This is because for higher values of $\varepsilon$ the privacy preserving noise becomes weaker due to which the contribution of the points in the minority groups are more effective when optimizing the KRR objective.

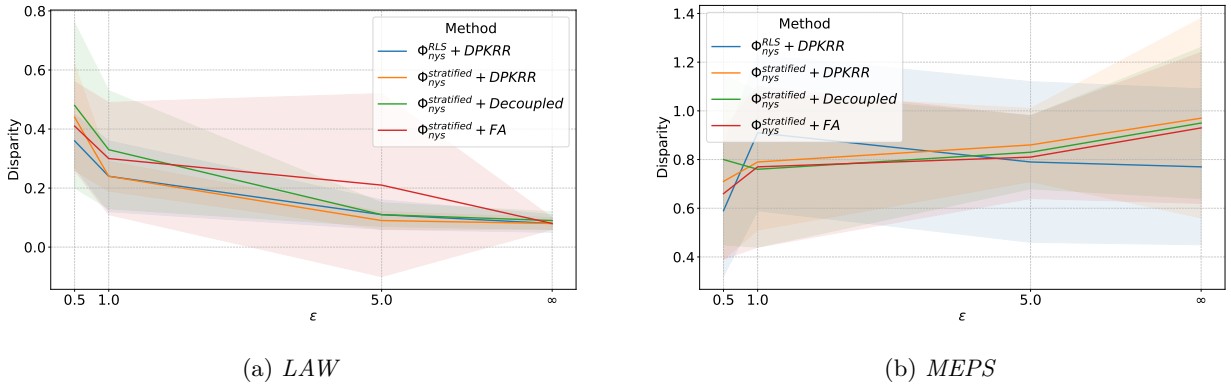

(a) *LAW*

(b) *MEPS*

Figure 3: Comparison of Maximum Disparity for $t = 150$.

However, for the *MEPS* dataset the disparities as well as the maximum group losses increase. One possible explanation for this is that when there is no privacy constraint, the KRR optimization learns better coefficients for one of the groups. The presence of privacy-preserving noise masks the differences between the groups. For the *LAW* dataset, since we ensure that the minority groups are allotted landmarks proportion-

ately to their leverage scores, all groups are well-represented. But when the groups are of similar sizes, both the groups are allocated similar number of landmarks. The coefficients learned in this case depend on how well the underlying patterns for each group can be learnt. The KRR minimization could minimize the loss for one group better, due to which the disparity increases.

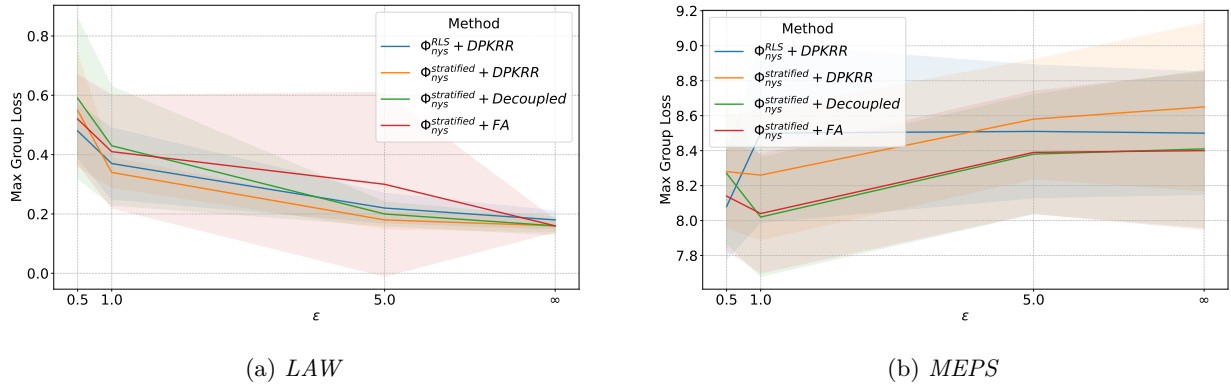

(a) *LAW*          (b) *MEPS*

Figure 4: Comparison of Maximum Group Loss for $t = 150$.

## 7 Conclusion and Future Work

In this work, we study the privacy and fairness issues in approximate kernel ridge regression. We propose a stratified RLS sampling approach that ensures that each group is allocated a proportional number of landmarks. Even though minimax fairness formulations require solving DRO-style optimization problems, for the private KRR optimization, we considered simple methods that have a closed form solution. We compared the fairness metrics across these methods. The empirical results highlighted the interplay between privacy and fairness, particularly the effect of group sizes. A theoretical analysis of this phenomenon is an interesting direction of future work, which could suggest the optimal number of landmarks to be sampled for a desired level of privacy and fairness. Another direction of future work is to have a minimax style formulation of the private KRR objective.

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
