# OpenReview forum: "Fair and Private Approximate Kernel Ridge Regression"
_TMLR — Withdrawn by Authors_

### Review · Reviewer_xFp2 · 2026-05-08

**Summary Of Contributions:**

This paper studies the Fair and Private ridge regression via the approximating kernels. The privacy is ensured by the differentially private KRR objective. Moreover, the minimax fairness is achieved through decoupling or feature augmentation. The experimental results show the performance of the considered algorithms on several datasets. The simultaneous applications of private and fair algorithms are novel, and the analyses and intuitions of fair algorithms look somewhat vague.

**Additional Comments:**

None

**Audience:**

Yes

**Audience Explanation:**

The combination of two issues can be interesting in the area of fields.

**Claims And Evidence:**

Yes

**Claims Explanation:**

The experiments are sufficient. However, the parts of fairness is too short.

**Requested Changes:**

The use of decoupling or augmented features is closely related to hierarchical modeling, often shown in statistical modeling. Usually, fairness can be achieved by stronger regularization of the more direct loss functions. Along with the studies of fairness, many algorithms of this type are considered. Therefore, the fair property obtained from the proposed looks less intuitive.

I want authors to present a more analytical discussion of fairness and results, examining how the performance discrepancy between groups changes as the regularizer term diminishes. Additionally, there is no detailed explanation concerning the shade in the plots. If it represents confidence intervals, there are too many overlaps, which degrade the statistical significance of the difference between algorithms.

---

### Review · Reviewer_YMot · 2026-05-12

**Summary Of Contributions:**

The paper studies the problem of scalable kernel ridge regression (KRR) under both fairness and differential privacy constraints. It proposes a framework based on Nyström approximation, where landmarks are selected using a stratified ridge leverage score (RLS) sampling strategy designed to ensure representation across demographic groups. The work further incorporates differentially private optimization of the approximate KRR objective using objective perturbation. The paper also explores two approaches for improving fairness: group-wise decoupled learning and feature augmentation.

The paper addresses an important and relevant problem at the intersection of scalable kernel methods, privacy, and fairness. The proposed stratified RLS sampling strategy is intuitive and empirically appears to improve disparity metrics compared to standard RLS sampling. The paper is generally well written and the empirical section provides some useful observations regarding the interaction between privacy noise, group imbalance, and approximation quality.

However, the work also has several limitations. The fairness objective is not directly optimized; instead, fairness is encouraged indirectly through the sampling strategy and modeling choices. The technical novelty is somewhat limited, as the framework mainly combines existing ingredients from Nyström approximation, leverage-score sampling, objective perturbation, and fair learning. In addition, the experimental evaluation is relatively limited in scale and scope, and the paper lacks theoretical guarantees regarding fairness, utility, or privacy-utility tradeoffs.

**Additional Comments:**

I think the paper addresses an important and interesting problem, and the proposed stratified landmark selection idea is promising. However, in its current form, the work feels somewhat preliminary for TMLR. The paper would be significantly strengthened by deeper theoretical analysis, stronger experimental validation, and a more precise framing of the fairness objectives being addressed.

**Audience:**

Yes

**Audience Explanation:**

The intersection of fairness, differential privacy, and scalable kernel learning is an important and relatively underexplored topic. Researchers interested in privacy-preserving machine learning, fair machine learning, kernel methods, and scalable learning algorithms may find the proposed framework and empirical observations useful.
In particular, the idea of incorporating group-aware landmark selection into Nyström approximation is interesting and could motivate further work on fairness-aware approximation methods for kernel learning. The paper also highlights practical interactions between privacy noise, group imbalance, and approximation quality that may be relevant for future research in this area.

**Broader Impact Concerns:**

The paper studies fairness and privacy in machine learning systems, which is societally important and potentially beneficial in sensitive domains such as healthcare. However, the work relies on demographic group information and assumes access to representative public data, which may not always hold in practice. The paper would benefit from a more detailed discussion of the limitations and risks associated with demographic data usage, potential distribution mismatch between public and private datasets, and the possibility that fairness improvements for some groups may not generalize across deployment settings.

**Claims And Evidence:**

No

**Claims Explanation:**

While the empirical results generally support the qualitative claim that stratified landmark sampling can improve disparity metrics compared to standard RLS sampling, I do not believe the evidence is fully sufficient to support several of the broader claims and framing of the paper.
In particular, the paper is framed as addressing “fair and private” KRR, including discussion of minimax fairness objectives. However, the proposed method does not directly optimize a fairness-constrained or minimax fairness objective; the paper explicitly leaves such formulations for future work. As a result, the fairness guarantees are largely heuristic and empirical rather than theoretically established.
The experimental evaluation is also somewhat limited. The paper evaluates only two relatively small datasets and compares against a narrow set of baselines. The work lacks stronger ablations, broader comparisons to existing fair or private learning approaches, and theoretical analysis of fairness, approximation quality, or privacy-utility tradeoffs. Several conclusions in the discussion section are speculative and not rigorously validated experimentally.
Overall, the paper provides preliminary empirical evidence supporting the usefulness of the proposed sampling strategy, but the evidence is not yet sufficiently comprehensive or rigorous to fully support the broader claims made in the paper.

**Requested Changes:**

1. Please clarify the paper’s fairness claims and framing. The current presentation suggests minimax or fairness-constrained optimization, but the proposed method does not directly optimize such objectives.

2. Strengthen the empirical evaluation with additional datasets, stronger baselines, and more comprehensive ablations (e.g., alternative landmark allocation strategies, sensitivity to public dataset quality, varying imbalance levels).

3. Please provide stronger theoretical analysis or guarantees regarding fairness, approximation quality, or privacy-utility tradeoffs.

4. Better justify the assumptions regarding the public dataset being representative of the private distribution and analyze robustness when this assumption does not hold.

5. Include runtime and scalability analysis compared to alternative approaches.

6. Improve discussion of the limitations of decoupled learning and feature augmentation approaches.

7. Add statistical significance analysis and confidence intervals where appropriate.

---

### Review · Reviewer_tuGn · 2026-05-13

**Summary Of Contributions:**

The paper looks at the intersection of three different problems concerning Kernel Ridge Regression (KRR): scalability, differential privacy, and group fairness. The authors aim to tackle these three aspects simultaneously, departing from Nystrom approximation method for scalable KRR. This is novel compared to existing related work. More specifically the contributions of the paper are the following:
+ A Fair and Private Kernel Ridge Regression (FPKRR) framework (in Algorithm 1) using a separate public dataset to sample the landmarks.
+ A stratified RLS sampling strategy for the Nystrom approximation that allocates the landmarks for different demographic groups proportionally to their collective contribution to their scores.
+ A differentially private KRR approach via objective perturbation.
+ Two methods for achieving fairness: 1) a decoupled approach and 2) a feature augmentation method.
The empirical evaluation is performed in two real-world datasets with different characteristics and compares 5 method variants using fairness metrics.

Strengths:
+ The interaction between fairness, privacy, and scalability is a very relevant topic, often underexplored, especially in the context of kernel methods. The paper provides a very interesting angle in this topic, and the motivation of the paper is clear and convincing.
+ The paper is well written and organized, especially in the first sections with the background and the related work. That helps with the positioning of the paper, its novelty and its contributions.
+ The integration of existing mechanisms to achieve scalability (via Nystrom method), RLS sampling, and objective perturbation into a reasonable pipeline. Despite its simplicity, Algorithm 1 provides a nice close-form solution avoiding solving more complex minimax optimization problems.
+ Consistent with prior work, the use of public data for the selection of the landmarks is smart and avoids some of the limitations of sampling from the training data (as the paper describes nicely) when using differential privacy (with some caveats that I discuss below).
+ The stratification strategy, allocating landmarks proportionally to leverage mass is an interesting approach and the motivation is clear.

Weaknesses:
+ Although the algorithms are well motivated, the paper lacks a theoretical analysis. For instance, the main algorithm has no formal guarantees. The paper acknowledges this is future work, but in related work in this area, this type of theoretical analysis is common to support the contributions.
+ The paper targets minimax group fairness, however, there is no explicit mechanism in the optimization objective itself that directly minimizes worst-group loss or disparity. This creates a disconnect between the stated goal and what the proposed method actually guarantees. In other words, it seems that reducing the group loss is a byproduct of the algorithm, not its main objective.
+ The full (ε, δ)-DP proof for the complete pipeline is missing, including the landmark selection step from the public dataset. Although the authors claim that using the public dataset preserves the privacy budget (Lowy et al. 2024), no formal composition arguments are provided.
+ The experimental evaluation is not strong: the authors only included two datasets and missed the inclusion of a large-scale dataset (despite the scalability motivation of the paper). The evaluation lacks the comparison with other baselines like, for example, non-private fair kernel methods, DP regression baselines without the Nystrom approximation, minimax fairness methods, or alternative landmark selection approaches. The baselines used in the paper are basically variations of the proposed method.
+ In the experiments the authors focus on reporting fairness metrics, but the results on the model’s utility are missing. Similarly, the results don’t report any statistical significance, which is important given the large error bars reported in some cases, and no broader fairness metrics are explored.
+ The experiments don’t include runtime or memory analysis, which is an important aspect given the focus on scalability.
+ The proposed approach relies on the access to a public dataset sampled from the same distribution and preserving demographic proportions. Although this is smart to bypass some of the limitations for enhancing privacy in this context, this is a strong assumption that may not hold in practice. In this sense, the paper lacks an analysis of the robustness of this assumption when, for example, the public and private distributions differ, minority groups are underrepresented in the public data, the public data is noisy, or how the size of the public dataset impacts the performance of the model.

**Audience:**

Yes

**Audience Explanation:**

The interdependence between scalability, privacy, and fairness is a very relevant angle in the design of modern machine learning pipelines. This interaction is often underexplored, especially in contexts like kernel methods.

**Claims And Evidence:**

No

**Claims Explanation:**

The claims made in the paper are partially supported by the evidence presented.
Some of the claims are supported empirically. For example, that using public data for landmark selection outperforms the DP clustering approach (Tables 1 and 2).
However, one of the central claims that stratified RLS sampling ensures fair group representation rests entirely on an intuitive argument. The algorithm does not explicitly aim to minimize group fairness, and the authors do not provide any formal guarantee. Also, the experimental support for this claim is also not convincing: the improvement over the standard RLS sampling appears to be meaningful on the LAW dataset, but the difference is unclear on a more balanced dataset like MEPS.
On the other hand, the privacy guarantees are not derived and formalized on the proposed algorithmic pipeline.
Finally, scalability is one of the angles that motivates the paper, however, the authors do not provide any empirical analysis on runtime measurements, memory analysis, scaling groups, or an experiment on a large-scale dataset.

**Requested Changes:**

+ Provide some formal guarantees, for example by proving a bound on the expected worst-group loss or disparity as a function of the privacy budget, group sizes, and number of landmarks.
+ Include a formal end-to-end (ε, δ)-DP analysis for Algorithm 1, including how composition is handled.
+ Clarify and moderate fairness claims as the current framing around minimax fairness is misleading as the proposed method does not aim to optimize this directly.
+ Improve the experimental evaluation: 1) stronger and meaningful baselines, 2) more datasets (including at least, one large-scale dataset), 3) add utility metrics, 4) add statistical significance tests, 5) ablation studies (how each component work on the pipeline), 6) sensitivity and robustness on the mismatch between the public and the private datasets, 7) Evaluate the methods under a wider range of ε values.

---

### Note · Authors · 2026-06-03

I have read and agree with the venue's withdrawal policy on behalf of myself and my co-authors.